# Generation of amine dehydrogenases with increased catalytic performance and substrate scope from ε-deaminating *L*-Lysine dehydrogenase

Vasilis Tseliou [1], Tanja Knaus [1], Marcelo F. Masman [1], Maria L. Corrado [1] & Francesco G. Mutti [1]

Amine dehydrogenases (AmDHs) catalyse the conversion of ketones into enantiomerically pure amines at the sole expense of ammonia and hydride source. Guided by structural information from computational models, we create AmDHs that can convert pharmaceutically relevant aromatic ketones with conversions up to quantitative and perfect chemical and optical purities. These AmDHs are created from an unconventional enzyme scaffold that apparently does not operate any asymmetric transformation in its natural reaction. Additionally, the best variant (LE-AmDH-v1) displays a unique substrate-dependent switch of enantioselectivity, affording *S*- or *R*-configured amine products with up to >99.9% enantiomeric excess. These findings are explained by in silico studies. LE-AmDH-v1 is highly thermostable ($T_m$ of 69 °C), retains almost entirely its catalytic activity upon incubation up to 50 °C for several days, and operates preferentially at 50 °C and pH 9.0. This study also demonstrates that product inhibition can be a critical factor in AmDH-catalysed reductive amination.

[1] Van 't Hoff Institute for Molecular Sciences, HIMS-Biocat, University of Amsterdam, Science Park 904, 1098 XH Amsterdam, The Netherlands. Correspondence and requests for materials should be addressed to T.K. (email: t.knaus@uva.nl) or to F.G.M. (email: f.mutti@uva.nl)

n 2007, when the Roundtable of the American Chemical Society's Green Chemistry Institute identified the most aspirational chemical transformations to challenge the pharmaceutical industry, the reductive amination of prochiral ketones with free ammonia ranked second on the list[1]. Indeed, α-chiral amines constitute approximately 40% of the optically active drugs that are currently commercialised mainly as single enantiomers[2]. These amines are classically synthesised industrially through the asymmetric hydrogenation of activated intermediates such as enamides, enamines, or pre-formed N-substituted imines[3]; however, such strategies lack efficiency because multiple chemical steps are required. Although significant progress in the asymmetric reductive amination of carbonyl-containing compounds has been recently achieved in organometallic catalysis (e.g., using Ir or Ru or Pd-based catalysts) and organocatalysis[4], a number of enzymatic routes offer more atom-efficient approaches, particularly for the synthesis of active pharmaceutical ingredients (APIs) in enantiomerically pure forms[5]. Furthermore, the often perfect stereoselectivity of enzymes eliminates the need for recrystallisation steps to upgrade the chemical purity of the enantiomer product, and there is no concern regarding the removal of traces of toxic heavy metals.

Classical industrially applied and laboratory scale biocatalytic methods convert α-chiral racemic amines into enantiomerically pure amines via either kinetic resolution (KR) or dynamic kinetic resolution (DKR) using hydrolases[6], or deracemisation using monoamine oxidases[7], the latter of which also gives access to optically active secondary and tertiary amines. Although KR is limited to a theoretical maximum of 50% yield, DKR and deracemisation require an additional chemical catalyst or a stoichiometric reagent[6,7]. Emerging enzymatic methods involve the use of ammonia lyases[8,9], Pictet-Spenglerases[10,11], or berberine bridge enzymes[12,13], all of which are active towards valuable types of intermediates for the synthesis of chiral amines; however, their substrate scope is rather restricted. Direct biocatalytic amination of unfunctionalised C-H bonds has been enabled by creating cytochrome P411, which requires tosyl azide as the nitrogen source[14]; however, this challenging strategy is limited to benzylic C-H amination and requires a subsequent deprotection step to remove the tosyl group. As such, asymmetric reductive amination of prochiral ketones and hydroamination of olefins are currently more atom-efficient methods for introducing an α-chiral amine moiety. However, the biocatalytic hydroamination of alkene moieties is restricted to the synthesis of α- and β-amino acids[8,9], whereas chemocatalytic hydroaminations yield preferentially secondary and tertiary amines with non-perfect enantiomeric excess[15]. The stereoselective conversion of ketones to amines can be accomplished either by a formal reductive amination using ω-transaminases[16] or a truly reductive amination using amine dehydrogenases (AmDHs), imine reductases (IReds) or reductive aminases (RedAms)[17,18]. IReds and RedAms are normally employed for the synthesis of secondary and tertiary amines[17,18], although this property was also recently observed in some cases with AmDHs[19]. Transaminases have proven to be useful and efficient biocatalysts for the stereoselective amination of ketones in laboratory, as well as industrial scale settings[16,20,21]; however, their downside is the requirement for supra-stoichiometric amounts of an amine donor and/or more enzymes and cofactors[16,20–22]. In this context, the AmDH-FDH system enables the efficient reductive amination of prochiral ketones (i.e., TONs > 10$^3$) while requiring only HCOONH$_4$ buffer and catalytic NAD[23]. A natural AmDH activity on aliphatic substrates was reported about two decades ago; however, the gene was never identified[24]. Native AmDHs were very recently identified for the reductive amination of aliphatic ketones with S-stereoselectivity

using a genome-mining approach[25,26]. Conversely, known engineered AmDHs have been obtained starting exclusively from L-amino acid dehydrogenases (deaminating at the α-amino group) and by mutating the same highly conserved amino acid residues in the active site (i.e., lysine and aspartate)[27–31]. Other AmDHs have been produced either by domain-shuffling or introducing further mutations into these first generation variants[32,33].

Accordingly, the diversity of substrate scope and activity among these AmDHs is poor because they were engineered from very similar scaffolds and following an identical strategy. For instance, all of the known AmDHs—whether native or engineered enzymes—exhibit mediocre or no activity towards the amination of acetophenones, tetralones, and chromanones, the related enantiomerically pure amines of which are recurrent motifs in many pharmaceuticals such as Sertraline, Norsertraline, Rotigotine, Rivastigmine, and Cinacalcet, among others[3].

Herein, we show that AmDHs can be engineered from a wild-type enzyme that apparently does not operate any asymmetric transformation. This property is revealed for an ε-deaminating L-lysine dehydrogenase from Geobacillus stearothermophilus (LysEDH), and it is exploited for the creation of AmDHs that perform the reductive amination of pharmaceutically relevant prochiral ketones with perfect stereo- and chemo-selectivity.

## Results

**Molecular modelling and preliminary activity studies.** The wild-type LysEDH catalyses the (formally) irreversible oxidative deamination of the ε-amino group of L-lysine (**1b**, Fig. 1a)[34]. This apparent irreversibility stems from the subsequent spontaneous cyclization of the ω-oxo amino acid product (**1a**). However, the reverse reaction must be possible if substrates devoid of the α-amine moiety (e.g., 6-oxo hexanoic acid) could be accepted by LysEDH (**2a**, Fig. 2b). Initial experiments on the reductive amination of **2a** (10 mM) in HCOONH$_4$/NH$_3$ (2 M, pH 9.0) revealed that LysEDH (90 μM) catalyses this reaction (44% analytical yield). The experiment was conducted in the presence of formate dehydrogenase (Cb-FDH) for NADH recycling (Fig. 2a) as described in Methods section (Biocatalytic reductive amination in analytical scale)[23,35]. Under the same conditions, substrates devoid of the ω-carboxyl moiety such as hexanal (**23a**), heptanal (**24a**), 2-heptanone (**5a**), 2-octanone (**25a**), acetophenone (**8a**) and α-chromanone (**13a**) were not converted (see Supplementary Fig. 1 for the structures of **23a**, **24a** and **25a**). Thus, the α-amino group (i.e., present in L-lysine) is not essential for productive substrate binding and amination catalysed by LysEDH, whereas the α-carboxyl group is crucial. These findings motivated us to undertake a structural examination of the active site architecture of LysEDH. Due to the lack of a crystal structure of LysEDH, a homology model was created based on published homologous structures. After several rounds of homology modelling, in silico modifications, and energy minimisation followed by molecular dynamic refinement, the model of LysEDH was obtained with both coenzyme (NADH) and ligand **1c** bound in the active site in the reactive pose (Fig. 1b, Supplementary Methods).

Our analysis of the model identified seven amino acid residues as potential targets for mutagenesis, which can be classified into two categories based on their role in the binding pocket (Fig. 1b). Group one encompasses residues that are appointed in the binding of the α-amino and α-carboxylic groups of **1b**, namely H181, Y238, and T240 and R242, respectively. These residues create a hydrophilic cavity that accommodates the hydrophilic α-amino acid moiety of **1b**. On the opposite side, residues F173, V172 and V130 create a hindered hydrophobic environment that forces the substrate into its ideal reactive pose and prevents its

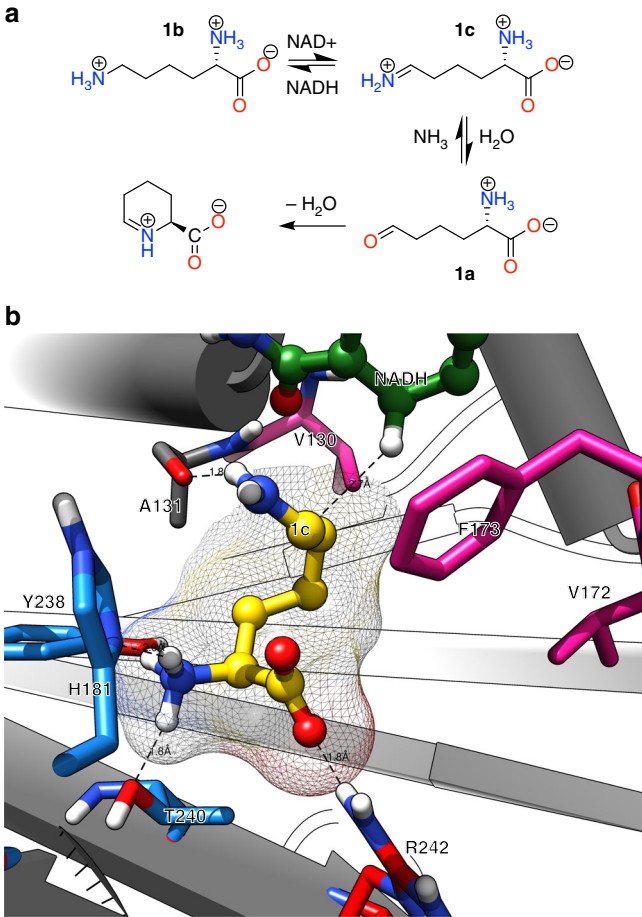

**a**

**b**

**Fig. 1** Model of the active site of the ε-deaminating L-lysine dehydrogenase from *G. stearothermophilus*. **a** Schematic representation of the reactions undergone by the natural substrate **1b**. **b** Model of the active site containing its natural substrate **1c**. Residues H181, Y238 and T240 (in cyan) appear to stabilise the alpha $NH_3^+$ group, whereas R242 appears to interact with the COO⁻ of the substrate. Residues V172, F173 and V130 create a hindered hydrophobic environment. All highlighted residues are amenable targets for protein engineering studies

rotation. In fact, in our model, the distance between the departing hydride of NADH and the pro-chiral carbon of the imine intermediate **1c** is 2.7 Å, which is below the sum of the van der Waals' radii of carbon and hydrogen atoms[36]. Since F173 is the bulkiest hydrophobic residue, we envisioned that its mutation into a less bulky residue such as alanine or glycine would retain the overall hydrophobic properties of the cavity while also permitting the binding of bulkier substrates in different orientations. Hence, ketones might be accepted besides simpler aldehydes.

As mentioned above, the wild-type LysEDH converted **2a** to **2b** with 44% analytical yield. As **2b** is a terminal (achiral) amine product, information on the reaction's stereoselectivity was not attainable; however, the reductive amination of **3a** as substrate (i.e., the structurally related methyl-ketone of **2a**) would result in an α-chiral amine. Since according to our model, the hydride of NADH must be transferred towards the *Si*-face of the prochiral carbonyl moiety, this experiment would reveal a possible stereoselectivity of LysEDH in the conversion of **3a**. Indeed, **3b** was produced in an enantiomerically pure *S*-configuration (**3a** 50 mM; LysEDH 90 μM; *ee* > 99%) and with 90% analytical yield. When LysEDH F173A (herein referred as LE-AmDH-v1) was tested under the same conditions, **3a** was aminated with > 99%

analytical yield and high stereoselectivity (*ee* > 99% *S*, Table 1). Moreover, preparative scale production of **3b** starting from **3a** (150 mg, 1.040 mmol), which aimed at identifying the product's absolute configuration, resulted in 73% isolated yield with *ee* > 99% *S* (Supplementary Methods).

**Asymmetric synthesis of amines catalysed by LysEDH variants.** The mutation F173A in the wild-type LysEDH created an additional large hydrophobic binding pocket in the active site, which might favour the binding of aromatic substrates such as **8a**. Analysis of our LysEDH model also evidenced that residues V172, V130 and R242 might be suitable targets for mutagenesis (Fig. 1b). Therefore, a focused library of 14 variants (LE-AmDH-v1 to v14, Fig. 2d) was created and screened against a number of aliphatic and aromatic ketones (Fig. 2b, group B, **4–16a**) in $HCOONH_4/NH_3$ buffer (pH 9.0, 2 M) using LE-AmDH variants (90 μM), Cb-FDH (19 μM), $NAD^+$ (1 mM) for 48 h at 30 °C. The screening outcome is summarised in Fig. 2c (with further details in Supplementary Tables 3–7). Notably, in a large majority of cases, the amine product was obtained in enantiomerically pure form (*ee* > 99%), albeit with the opposite configuration (*R*) than that observed for the natural binding mode of *L*-lysine (pro-*S*).

The mutation F173G (v2) resulted in a significant loss of activity compared to the beneficial F173A (v1). Compared with LE-AmDH-v1, LE-AmDH-v2 showed an approximately three-fold decrease of conversion values for the amination of **8a** and **9a** and a two-fold decrease for the amination of **10a**. Moreover, LE-AmDH-v2 did not convert **4a**, **5a**, **12a**, **15a** and **16a** into the corresponding amines, whereas LE-AmDH-v1 yielded, respectively, 78%, 10%, 61%, 26% and 58% conversion for the same substrates. All variants bearing the mutation F173G (LE-AmDH-v2, v3 and v4) generally showed very low levels of activity towards the tested substrates, thus indicating that an increase of flexibility in the active site is detrimental. Substitution of the neighbouring V172 either to alanine or glycine while maintaining the beneficial F173A mutation (LE-AmDH-v5 and v6) provoked a dramatic reduction of catalytic activity. Interestingly, the residue R242 appears to be involved in the binding of the α-carboxylic group of *L*-lysine and it is located at the entrance of the active site in a region that is exposed to an aqueous environment (Supplementary Fig. 4). To investigate if R242 regulates accessibility of substrates to the active site, we also created the variants R242M (v14), R242M/F173A (v7), R242M/F173V (v13) and R242W/F173A (v8). LE-AmDH-v7 converted substrates **8a** (6%) and **9a** (19%, *ee* > 99% *R*), whereas LE-AmDH-v13 converted **8–10a** and **13a** (38%, 41%, 8%, 15% conversions, *ee* > 99% *R*, Supplementary Table 4–6), thus indicating that increasing the hydrophobicity of the binding pocket at this site, with minimal conformational rearrangement, reduces enzyme activity.

Finally, we investigated if the mutation of V130, which is located on the opposite face of F173A, to a less bulky hydrophobic amino acid residue such as alanine or glycine, would enable a different binding mode for the substrate. LE-AmDH-v1 (F173A) was combined with mutations V130A or V130G to generate LE-AmDH-v11 and v12. LE-AmDH-v11 exhibited a significant level of activity towards **8a** (74% conversion) and **9a** (65% conversion), and all amine products (**8–13b**) were obtained in enantiomerically pure form (*ee* > 99% *R*). LE-AmDH-v12 was also stereospecific (*ee* > 99% *R*) for the amination of **9a** (78% conversion) and other ketones (**8–9a**, **12–13a**, **16a**, Supplementary Tables 4–7). Nonetheless, LE-AmDH-v12 also aminated **4a** and **15a** (64 and 42% conversion), albeit with imperfect stereoselectivity (*ee* 83% *R* and 77% *R*). These results indicate that a productive pro-*S* binding pose was generated in these two reactions in addition to the favoured pro-

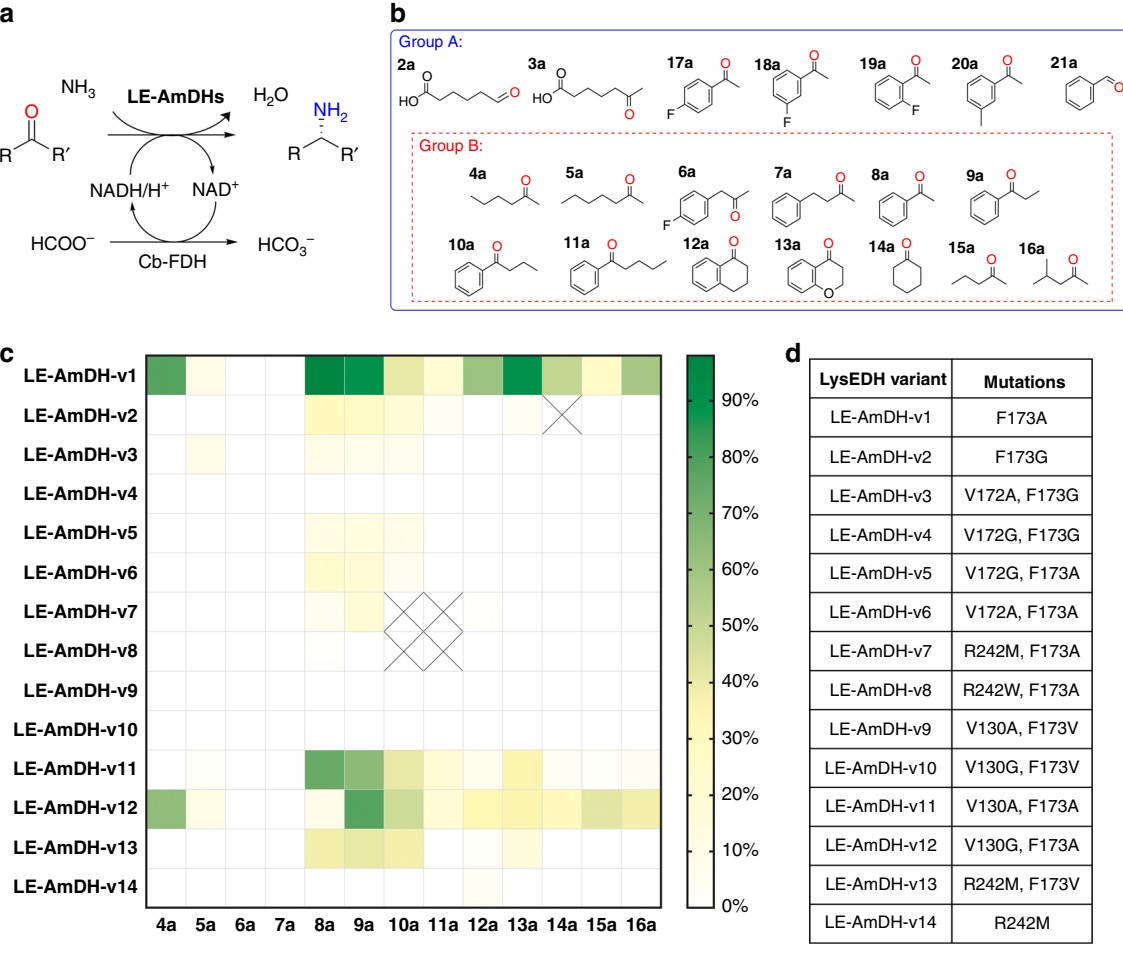

**Fig. 2** Initial screening with LE-AmDH variants. **a** General scheme of the biocatalytic reductive amination performed by LE-AmDH variants. A catalytic amount of NAD$^+$ (1 mM) was applied. The reducing equivalents, as well as the nitrogen source originated from the buffer of the reaction: HCOONH$_4$/NH$_3$ (pH 9.0, 2 M). The substrate concentration was 10 mM. AmDH and Cb-FDH were used in final concentrations of 90 μM and 19 μM, respectively. Reaction volume: 0.5 mL; reaction time: 48 h; temperature: 30 °C; agitation on an orbital shaker at 170 rpm. **b** Group B: substrates used for screening of the LysEDH variants reported here. Group A: additional substrates tested with LE-AmDH-v1. **c** Heatmap of the screening outcome, in which Group B substrates were tested to reduce screening effort (results expressed in conversion). **d** List of mutations introduced in the LysEDH scaffold

*R*. Architectural modulation of the active site by testing the F173V mutation generated inactive variants (LE-AmDH-v9 and v10). Figure 2c shows that the best variant for the reductive amination of the target aromatic ketones turned out to be LE-AmDH-v1. Accordingly, further studies in this work were conducted with that AmDH.

**Biocatalytic optimisation studies on LE-AmDH-v1.** The temperature dependence and influence of the pH value were investigated for the reductive amination catalysed by LE-AmDH-v1 using **8a** as test substrate. As it originates from the thermophile bacterium *Geobacillus stearothermophilus*, the wild-type LysEDH displays high stability and activity at elevated temperatures[34]. The same features were observed for the LE-AmDH-v1 variant. Figure 3a (and Supplementary Table 8) shows that conversions of **8a** were above 90% after 16 h for reactions performed in the range of 30–60 °C. Remarkably, at 50 °C and 60 °C, conversions were found to already be 80 and 90%, respectively, after the first 4 h. Conversely, the conversion at 20 °C did not exceed 90% even after 32 h. A subsequent evaluation of the influence of buffers (HCOONH$_4$/NH$_3$ 2 M) at different pH values revealed the highest conversions at pH 9.0–9.5 (Fig. 3b, Supplementary Table 9).

The thermal robustness of LE-AmDH-v1 motivated us to determine its thermostability data at different pH values using HCOONH$_4$/NH$_3$ buffer (2 M, pH 7.0–9.5) in the presence, as well as absence of ligand **8a** and coenzyme (NADH). In all cases (Supplementary Methods and Supplementary Table 17), the $T_m$ was found to be 65 °C, which indicates no influence of pH in the thermal stability of LE-AmDH-v1. Notably, $T_m$ was increased to 69 °C when NADH was present in the mixture. All of these data align with the thermal stability profile of the wild-type LysEDH[34], thus indicating that the mutation F173A did not affect the stability of LE-AmDH-v1.

Finally, the long-term stability of LE-AmDH-v1 was tested in HCOONH$_4$/NH$_3$ (2 M) buffer at pH 9.0 at different temperatures and using various batches of enzyme for a maximum of 7 days (Supplementary Methods). Remarkably, no loss of activity was observed within this time period for samples incubated at 4 °C and at room temperature. Samples incubated at 40 °C exhibited a consistent residual activity between 80 and 90% after 7 days, whereas samples incubated at 50 °C showed an average residual activity of ca. 60% and a maximum residual activity of 80% after 7 days. Notably, the samples incubated at 60 °C (close to the $T_m$) still showed a residual activity of ca. 50% after 70 hours. These data further support the real applicability of LE-AmDH-v1 in organic synthesis.

**Table 1 Reductive amination of a panel of ketones and aldehydes employing LE-AmDH-v1**

| Substrate | 10 mM (Conv. %) | 50 mM (Conv. %) | ee (%)[a] |
|---|---|---|---|
| 2a | >99 | >99 | N.A. |
| 3a | >99 | >99 | >99 (S)[b] |
| 4a | 87 | 71 | 99 (R) |
| 8a | >99 | 98 | >99.9 (R)[b] |
| 9a | 98 | 95 | >99.9 (R)[b] |
| 10a | 87 | 65 | 99.8 (R) |
| 11a | 36 | 26 | N.D. |
| 12a | 79 | 50 | >99 (R)[b] |
| 13a | 82 | 55 | 99.4 (R) |
| 14a | 86 | 84 | N.A. |
| 15a | 86 | 81 | 89 (R) |
| 16a | 76 | 53 | 97 (R) |
| 17a | 99 | 92 | >99.8 (R)[b] |
| 18a | 99 | 96 | >99.7 (R)[b] |
| 19a | >99 | 98 | >99.8 (R)[b] |
| 20a | 88 | 73 | >99 (R)[b] |
| 21a | >99[c] | >99[c] | N.A. |

Reaction conditions: substrate (10 mM or 50 mM), LE-AmDH-v1 (90 μM), Cb-FDH (19 μM), NAD$^+$ (1 mM), HCOONH$_4$/NH$_3$ buffer (2 M, pH 9.0), T 50 °C, agitation orbital shaker 170 rpm, reaction time 48 h

*ND* could not be determined, *NA* not applicable

[a]In some analytical scale reactions, it was impossible to determine the *ee* values with an accuracy equal or superior to 99.7% due to the elevated conversion and the high response GC factor of the amine product. The 99.7% *ee* threshold value is important, as it is set by law for the manufacture and commercialisation of APIs (i.e., max 0.15% total impurities, with the minor enantiomer being measured as impurity). High stereoselectivity was also confirmed in the preparative scale reactions

[b]In these cases, the level of precision depends on instrumental detection limits rather than the intrinsic stereoselectivity of the enzyme. Indeed, the *S*-enantiomer was never observed.

[c]Reaction in HCOONH$_4$/NH$_3$ buffer (2 M, pH 7.8)

**Substrate scope of LE-AmDH-v1**. LE-AmDH-v1 exhibited a significantly extended and complementary substrate scope for aromatic substrates compared with the currently available AmDHs (Table 1). Specifically, Rs-AmDH, Bb-AmDH and Cal-AmDH can efficiently aminate 4-phenylbutan-2-one (and derivatives) and/or phenylacetone (and derivatives)[23,28,29,31], whereas other AmDHs such as Ch1-AmDH, LeuDH, AmDH4, MsmeAmDH, MicroAmDH, CfusAmDH and EsLeuDH enzymes exhibit particularly high activity towards aliphatic substrates[23,25,27,30,33,37]. In contrast, elevated biocatalyst loading of the previously described Ch1-AmDH (130 μM) poorly aminated a 50 mM solution of **8a** (34% conversion), derivatives thereof (**17–20a**, 9–43% conversion), and **9a** (8% conversion) within 48 h[23]. Bulky-bulky ketones such as **10a** and **11a** were not accepted at all by Ch1-AmDH[23]. In another case, an elevated concentration of an AmDH engineered from *Exigobacterium sibiricum* (EsLeuDH, 1.45 mM; concentration calculated from a reportedly amount of 10 U mL$^{-1}$, activity of 0.17 U mg$_{enzyme}^{-1}$ and MW$_{enzyme}$ 40.5 kDa) was necessary to reach ca. 80% conversion of **8a** in 100 h[38]. In contrast, LE-AmDH-v1 (90 μM) converted the above-mentioned acetophenone and derivatives thereof (**8a** and **17–20a**, 50 mM) with conversions up to 98% within 48 h (Table 1). Propiophenone (**9a**) was accepted (50 mM, 95% conversion), as well as the more bulky substrates **10a** (50 mM, 65%) and **11a** (50 mM, 26%). Moreover, neither Ch1-AmDH[23,32] nor other previously reported AmDHs (e.g., Cal-AmDH)[23,25,27–31,33,37] are capable of aminating either α-tetralone (**12a**) or α-chromanone (**13a**) to any synthetically useful extent. In contrast, LE-AmDH-v1 exhibited conversions above 50% for the amination of these substrates (50 mM, Table 1). In addition, LE-AmDH-v1 proved to be a versatile biocatalyst, such that aliphatic ketones (**4a**, **14–16a**, 50 mM) were also accepted (53–84% conversion) and benzaldehyde (**21a**, 50 mM) was quantitatively aminated. Lastly, LE-AmDH-v1 is a high stereoselective catalyst

that produces either *S*-configured ω-amino acid **3a** or *R*-configured α-chiral amines in elevated enantiomeric excess (Table 1).

**Reaction intensification**. Using the optimum pH value (pH 9.0), a panel of ketones (**8a**, **9a**, **13a**, **17–19a** and **21a**) was tested for reductive amination from 10 mM up to 100 mM final substrate concentrations at two different temperatures (30 °C and 50 °C), since higher temperatures may affect the stability of the enzyme under process reaction conditions (e.g., shaking). The conversions and productivities are displayed in Fig. 3c to i (also see Supplementary Tables 10, 11). The highest productivity was observed at 50 °C and 100 mM substrate concentration for aminations of **8a** (84 mM), **9a** (89 mM) and **18a** (58 mM). The highest productivity for amination of **17a** (at 50 °C) was observed with 50 mM of substrate concentration resulting in 46 mM of the corresponding amine product. LE-AmDH-v1 could also produce 69.6 mM of **19b** (at 50 °C) and 38.3 mM of **13b** (at 30 °C) starting from 75 mM of **19a** and **13a**, respectively. Finally, the highest activity was observed with benzaldehyde (**21a**), which was fully converted by LE-AmDH-v1 at 200 mM scale at 50 °C.

**Reductive amination reactions in preparative scale**. In order to confirm the potential usefulness of LE-AmDH-v1 for the manufacturing of APIs, we performed preparative scale reductive amination reactions (100 mL reaction volume) starting from **8a** (600 mg, 5 mmol) and **9a** (671 mg, 5 mmol), which were converted into (*R*)-**8b** and (*R*)-**9b** with >99 and 97% conversion, respectively. The unreacted ketone substrate was removed by extraction with MTBE under acidic conditions. After basification, the amine products (*R*)-**8b** and (*R*)-**9b** were extracted with MTBE in 82 and 65% isolated yields, respectively. No further purification was required. By considering the detection limit of the GC measurement and concentration of the analytical samples (Supplementary Methods), we could conclude that (*R*)-**8b** and (*R*)-**9b** were obtained in >99.9% *ee* and total impurities were below the detection limit (i.e., surely <0.15%); these values comply with the requirement for API commercialisation.

**Steady state kinetic data and inhibition studies**. LE-AmDH-v1 was characterised with respect to its kinetic properties for the amination of **8a** and **21a**. First, steady-state kinetics were determined for the amination of **8a** at different temperatures; the most elevated catalytic profile was observed at 60 °C (Table 2, Supplementary Methods and Supplementary Table 19). Therefore, all other enzymatic assays were performed at 60 °C. LE-AmDH-v1 exhibited both the highest $k_{cat}$ value and the lowest $K_M$ for **21a** (Table 2, Supplementary Methods and Supplementary Table 19).

With the aim of rationalising the higher conversion of **8a** into **8b** that LE-AmDH-v1 exhibits over Ch1-AmDH, we also determined the kinetic parameters of Ch1-AmDH for the reductive amination of **8a** (Supplementary Methods and Supplementary Table 19). Surprisingly, despite the much higher conversion for the amination of **8a** (50 mM) catalysed by LE-AmDH-v1 (>99%, 90 μM enzyme) compared with Ch1-AmDH (34%, 130 μM enzyme) after 48 h[23], Ch1-AmDH possesses ca. two-fold higher $k_{cat}$ and similar $K_M$ values of those of LE-AmDH-v1 for the amination of **8a** at 60 °C (Table 2). Therefore, we hypothesised that the reductive amination of **8a** catalysed by Ch1-AmDH could be affected by severe product inhibition, as has also been observed with other aminating enzymes such as ω-transaminases[16,39,40]. We remark that steady-state $k_{cat}$ and $K_M$ values are commonly determined at an initial zero concentration of the product of the reaction. As such, solely considering these parameters for the evaluation of the catalytic effectiveness of a

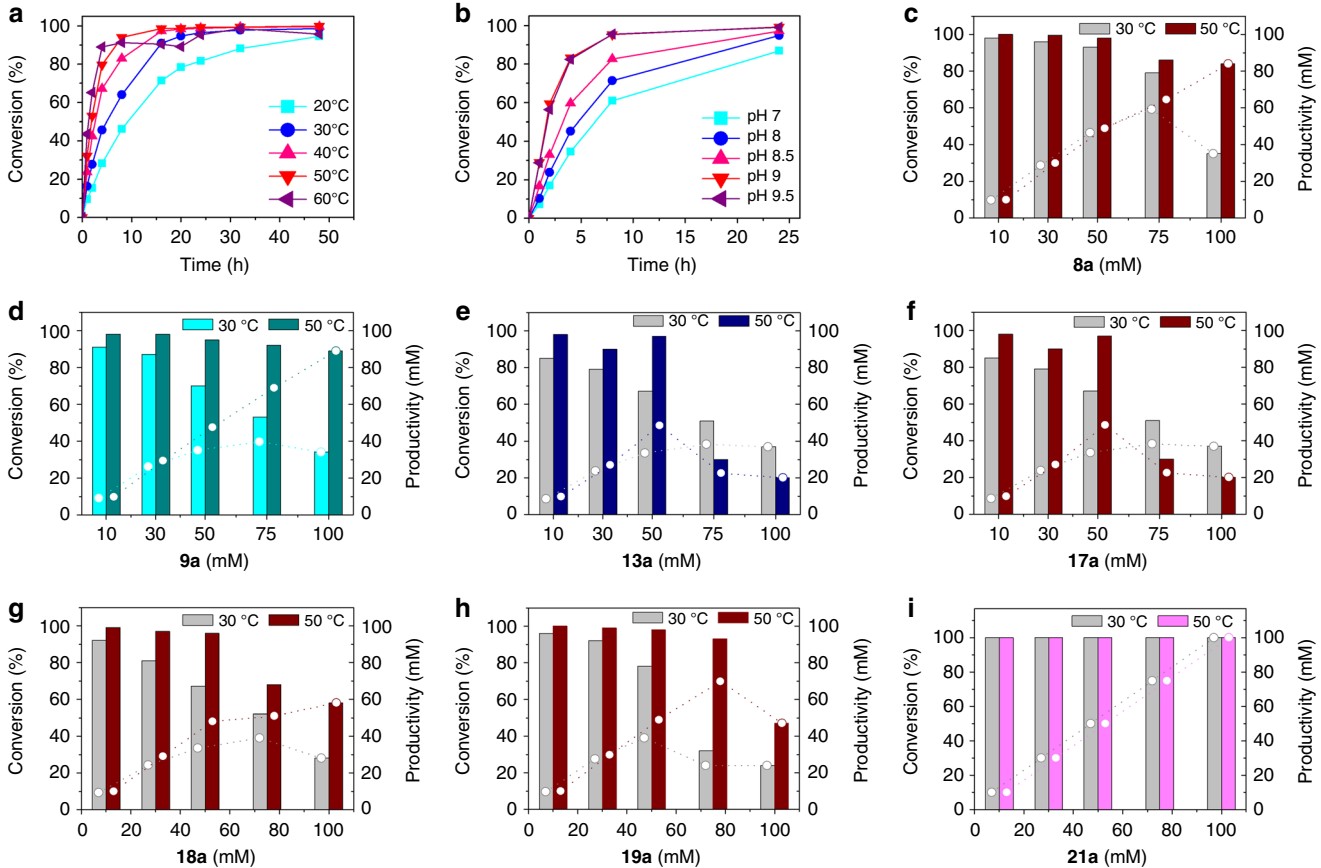

**Fig. 3** Optimisation studies employing LE-AmDH-v1. Reaction conditions: 0.5 mL final volume; buffer: HCOONH$_4$/NH$_3$ 2 M pH 7.0–9.5; 170 rpm on orbital shaker; [substrate] = 10–100 mM; [NAD$^+$] = 1 mM; [LE-AmDH-v1] = 90 µM; [Cb-FDH] = 19 µM; reaction time: 1–48 h (**a–b**) and 48 h (**c–i**) **a** Influence of the temperature (20 to 60 °C) on the reductive amination of **8a** in HCOONH$_4$/NH$_3$ 2 M pH 9.0 buffer. **b** Influence of the pH values using HCOONH$_4$/NH$_3$ buffer at 50 °C on the reductive amination of **8a**. **c–i** Effect of the substrate concentration on conversions (left y-axis; columns) and productivities (right y-axis; dashed lines with symbols) at two different temperatures: 30 and 50 °C. Source data are provided as a source data file. Data are based on single measurements

biocatalyst can lead to wrong conclusions, particularly when product inhibition occurs, as reviewed elsewhere[41,42].

Initially, we studied the potential inhibitory effect of the amine product (R)-**8b** on LE-AmDH-v1 and Ch1-AmDH at a 15 mM concentration of **8a** (Supplementary Methods and Supplementary Table 20) by determining the IC$_{50}$ values, which were indeed 20 mM for LE-AmDH-v1 and 1 mM for Ch1-AmDH (Fig. 4b and Table 2). A subsequent set of in-depth kinetic experiments on competitive product inhibition followed by Lineweaver-Burk analysis (Fig. 4c–e, Supplementary Methods and Supplementary Tables 21–22)[43] revealed that Ch1-AmDH is ca. 38-fold more inhibited than LE-AmDH-v1 (Table 2, see $K_I$ values). The evaluation of the overall catalytic performance ($k_{app}/K_M^{eff}$) according to Fox et al.[41], demonstrated that LE-AmDH-v1 is ca. 15-fold more efficient than Ch1-AmDH (Table 2, Supplementary Methods and Supplementary Table 23). In this evaluation, $K_M^{eff}$ acts as the time-averaged and effective value for the $K_M$ over the course of the reaction, thus including the competitive product inhibition phenomenon. Notably, an extremely high value of $K_M^{eff}$ for the conversion of **8a** with Ch1-AmDH indicates that the theoretical v$_{max}$ (i.e., $k_{app}$)—although mathematically higher—already becomes physically unattainable at the beginning of the reaction. In conclusion, our kinetics explain the superior catalytic performance of LE-AmDH-v1 in the reductive amination of **8a**. Notably, LE-AmDH-v1 is also capable of converting prochiral ketones that are not accepted by other AmDHs at all.

**Computational studies**. The final refined model of LE-AmDH-v1 (Fig. 1b, Supplementary Methods and Supplementary Table 12) was used for in-depth computational studies aimed at explaining the stereoselective outcome of the reactions. We have shown that linear or aromatic ketones (e.g., **4a**, **8a**) can be quantitatively aminated with perfect stereoselectivity (ee > 99% R). Conversely, the amination of **3a** proceeded quantitatively with opposite stereoselectivity (ee > 99% S). A substrate-dependent switch of the enantioselectivity enabled by introducing a single mutation (as the F173A in LE-AmDH-v1) has been observed in very rare cases in enzyme catalysis such as with an ene-reductase[44], a naphthalene dioxygenase[45], a lipase[46], and an ω-transaminase[47]. However, a substrate-dependent switch of enantioselectivity has been observed only in the latter two cases, albeit the resulting enantiomeric excess was poor (ca. 58%)[46,47]. Therefore, LE-AmDH-v1 is the first example of a biocatalyst possessing substrate-dependent stereo-switchable selectivity while always affording perfect enantiopurity of the products. Other examples from the literature describe either more mutations and consequent major structural alterations[48], or starting from scaffolds with poor selectivity[49,50], or changing the reaction medium[51]. To explain this singular feature, the iminium intermediates **4c**, **8c** and **3c** were docked into the active site of the enzyme. Via molecular docking and molecular dynamics simulations, we were able to identify the putative productive pro-R and pro-S binding modes for all three iminium intermediates. The two parameters to consider are[19]: (i) the distance between the departing hydride of

**Table 2 Steady state kinetic data and inhibitory effects obtained for LE-AmDH-v1 and Ch1-AmDH[a]**

| Enzyme | Substrate | T (°C) | $k_{app}$ (min$^{-1}$) | $K_M$ (mM) | IC$_{50}$[15] (mM)[b] | $K_I$ (mM)[c] | $K_M$$^{eff}$ (mM)[c] | $k_{app}/K_M$$^{eff}$ (M min$^{-1}$) |
|---|---|---|---|---|---|---|---|---|
| LE-AmDH-v1 | **8a** | 60 | 7.1 ± 0.2 | 5.5 ± 0.5 | 20 | 4.480 | 467 | 15 |
| | | 50 | 7.0 ± 0.1 | 7.2 ± 0.4 | | | | |
| | | 40 | 5.7 ± 0.1 | 9.6 ± 0.6 | | | | |
| Ch1-AmDH[a] | **8a** | 60 | 17.7 ± 0.4 | 5.6 ± 0.4 | 1 | 0.120 | 16838 | 1 |
| | | 50 | 13.1 ± 0.5 | 8.5 ± 0.8 | | | | |
| | | 40 | 7.7 ± 0.3 | 11.6 ± 0.9 | | | | |
| LE-AmDH-v1 | **21a** | 60 | 43.6 ± 1.7 | 4.9 ± 0.5 | | | | |

[a]The determination of the catalytic parameters of Ch1-AmDH by Bommarius and coworkers (ref. [32]) are very similar to those obtained in this study: $k_{app}$ 14.4 min$^{-1}$ (60 °C), 10.8 min$^{-1}$ (50 °C) and 7.2 min$^{-1}$ (40 °C); $K_M$ 5.2 mM (60 °C). Source data are provided as a source data file. ± represent the deviation ($n = 2$ independent experiments using two different enzyme batches)
[b]Determined at a concentration of 15 mM **8a**
[c](R)-**8b** was used as inhibitor. Following Fox et al. (ref. [41]), $K_M$$^{eff}$ was calculated based on the determined $K_I$, [S]$_0$ = 100 mM and c = 99% (for further details, see Supplementary Methods)

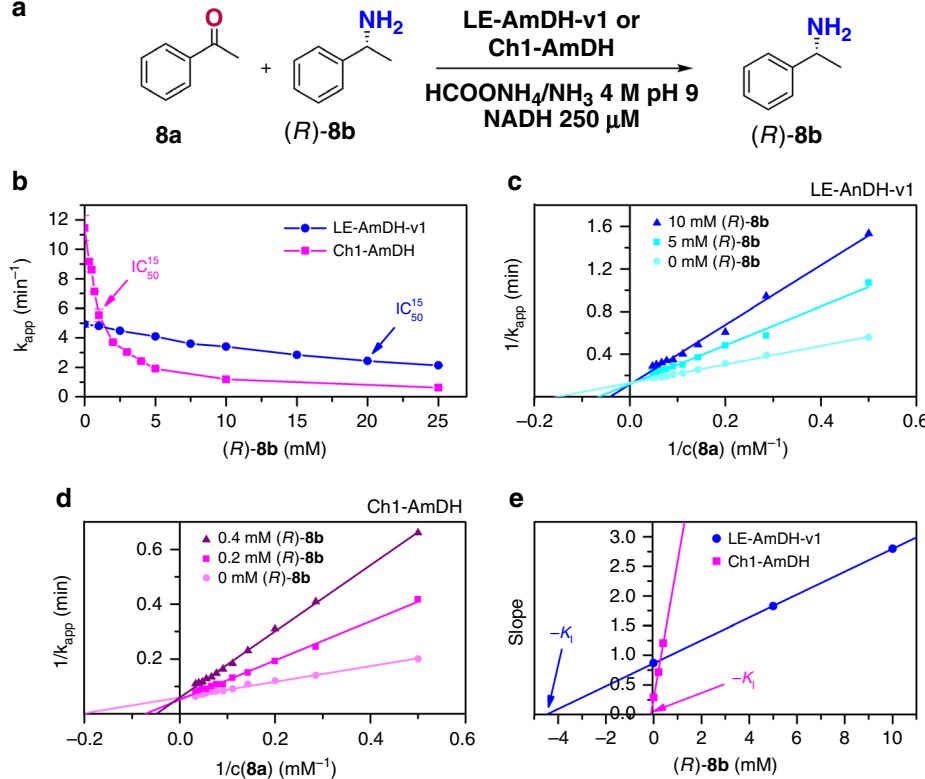

**Fig. 4** Product inhibition studies employing LE-AmDH-v1 and Ch1-AmDH. **a** Overall scheme of the kinetic experiments, in which **8a** is used as substrate and (R)-**8b** as competitive inhibitor. **b** Inhibitory effect (IC$_{50}$[15]) of the amine product (R)-**8b** at 15 mM of substrate concentration **8a**. **c** and **d** Primary double reciprocal plots for competitive inhibition according to Lineweaver-Burk analysis for LE-AmDH-v1 and Ch1-AmDH, respectively. **e** Secondary plots of the slopes obtained from the primary Lineweaver-Burk plots vs. inhibitor concentrations. The absolute value of the intercept with the x-axis provides the competitive inhibition constant ($K_I$). Source data are provided as a source data file. Data points of Fig. 4b are the average of $n = 2$ independent measurements, but the deviation is so low that the error bar is smaller than the dots; the only visible error bar is for the pink line at $k_{app}$ca. 5. Figure 4c, d are based on single measurements. Figure 4e is a secondary plot derived from Fig. 4c, d

NADH and *pro-chiral* carbon of the iminium intermediate (Supplementary Methods); and (ii) the behaviour of a dihedral angle as defined by the three atoms bonded to the *pro-chiral* carbon of the iminium intermediate and the hydride atom of the NADH. This angle determines the optical configuration of the amine product (for details, see Methods section and Supplementary Methods). The data are summarised in Fig. 5a.

In the case of the iminium intermediate (**8c**) generated from **8a**, the *pro-R* conformation is highly favoured with 30.5% of the MD snapshot (within the simulation time) below the threshold distance of 3 Å between the hydride of NADH and prochiral

carbon of **8c**. Starting from the *pro-S* conformation of **8c**, only 0.6% of MD snapshot is favourable for reduction. The same scenario occurs for the iminium intermediate **4c** (generated from **4a**). As expected, the situation is reversed for the reduction of **3c** (generated from **3a**). In this case, the *pro-S* conformation assumes a value below the threshold distance at 70.6% of the simulation time compared with at only 4.30% of the time for the *pro-R*.

Figure 5b represents a superposition of the structure of the wild-type LysEDH and the LE-AmDH-v1 (i.e., F173A variant) with bound iminiums of *L*-lysine (**1c**) and acetophenone (**8c**).

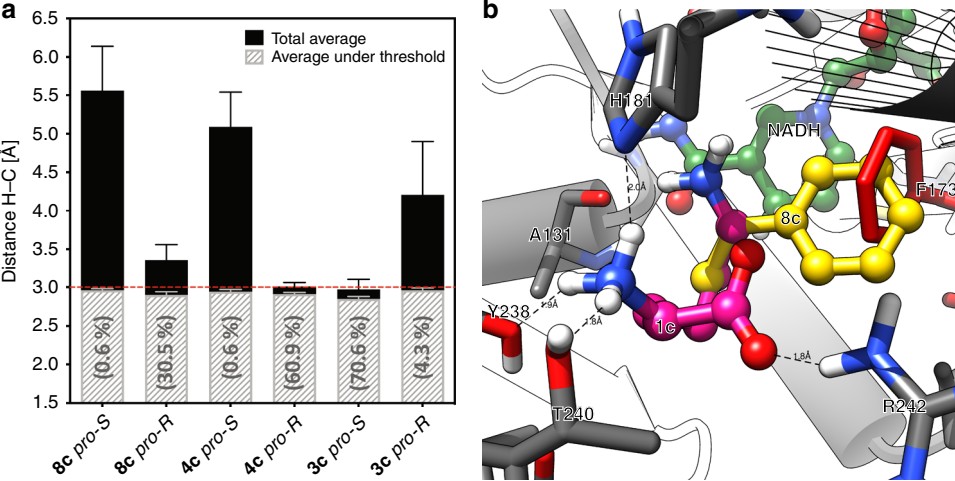

**Fig. 5** Pro-chiral preferences of LE-AmDH-v1. **a** Total average hydride/*pro*-chiral carbon distance over a minimum of 6 MD simulations and average under the distance threshold (3 Å). The percentiles indicate how often the average distance was under the given threshold. For details, see Supplementary Methods. **b** Superposition of the natural substrate and **8a** in the active site of the *wild type* enzyme. The natural intermediate **1c** is depicted in magenta, whereas the iminium of acetophenone (**8c**) is depicted in yellow. The mutation point (F173A) is marked in red and the cofactor NADH is depicted in green. It can be observed that once the mutation F173A is introduced, the aromatic ring of **8c** gets accommodated into the newly created hydrophobic cavity, which was previously occupied by the phenyl group of F173; that produces an inversion of the chiral preferences of this mutant towards aromatic substrates and towards substrates with (relatively) short hydrophobic chains. Source data are provided as a source data file. Error bars represent the standard deviation of *n* = 6 independent experiments

Ligand **8c** is oriented to have its phenyl ring in the new cavity created by the mutation F173A. This binding is not possible in the wild-type enzyme due to a steric clash with the side chain of F173. In contrast, **1c** always assumes the "natural" conformation as reported in the initial homology model (Fig. 1b).

## Discussion
In summary, guided by structural information obtained from computational studies, we have generated AmDHs starting from an enzyme that does not operate any apparent asymmetric transformation in its natural reaction. The best variant (LE-AmDH-v1) is highly thermostable ($T_m$: 69 °C), also exhibiting highly retained catalytic activity (>99% at RT, 90% at 40 °C and 80% at 50 °C) upon incubation for 7 days. LE-AmDH-v1 operates preferentially at 50 °C and pH 9.0, thus affording pharmaceutically relevant amines such as **8b** (and derivatives), **12b**, **13b** and **9b** in enantiomerically pure *R* configuration and elevated productivities. Such biocatalytic performance is in part due to a remarkably reduced product inhibition, as investigated for the amination of **8a**. LE-AmDH-v1 also quantitatively aminated **3a** into an *S* configured product (*ee* > 99%), thus providing a rare example of substrate-dependent stereo-switchable selectivity. Finally, in silico studies provided insight into the role of the mutations upon ligand binding and explained the enantioselective preferences of LE-AmDH-v1.

## Methods
**Biocatalytic reductive amination in analytical scale**. SDS-page documenting the purity of the AmDHs is reported in Supplementary Methods. LysEDH variant (90 μM, 4 mg mL⁻¹), NAD⁺ free acid (1 mM), formate dehydrogenase from *Candida boidinii* (Cb-FDH, 19 μM, 0.81 mg mL⁻¹) and substrate (**4–21a**, 10–200 mM) were added to an ammonium formate buffer (HCOONH₄/NH₃ 0.5 M, 2 M, pH 9). Biotransformations were performed at various temperatures and for different lengths of time in Eppendorf tubes in a horizontal position on orbital shakers (170 rpm). The reactions were quenched by the addition of aqueous 10 M KOH (100 μL) and extracted with dichloromethane (1 × 600 μL). After centrifugation, the organic phase was dried with MgSO₄ and the conversion was measured by GC-FID. Prior to injection into a chiral column for the determination of the *ee*, derivatisation of the samples was performed by the addition of 4-

dimethylaminopyridine in acetic anhydride (50 μL of stock solution 50 mg mL⁻¹, 409 mM). The samples were shaken in an incubator at RT for 30 min, after which water (500 μL) was added and the samples were shaken for an additional 30 min. After centrifugation, the organic layer was dried with MgSO₄. Enantiomeric excess was determined by GC with a variant Chiracel DEX-CB column. Details on the GC analysis and methods are reported in Supplementary Figs. 5–18 and Supplementary Tables 1 and 2. Analytical reference compounds were synthesised as reported in Supplementary Methods and Supplementary Tables 14–16 or purchased as reported in Supplementary Notes.

In the cases of **2a** and **3a**, the reactions were quenched by the addition of 100 μL HCl (3 M) and the samples were centrifuged (15 min, 21.1 g). The supernatants were analysed by RP-HPLC (Supplementary Methods)

**Biocatalytic reductive amination in preparative scale**. Preparative scale reactions of **8a** (50 mM, 600 mg, 5 mmol) and **9a** (50 mM, 671 mg, 5 mmol) were performed in a total volume of 100 mL of HCOONH₄/NH₃ buffer (2 M, pH 9.0) at 50 °C in 250 mL Erlenmeyer flasks containing LE-AmDH-v1 (90 μM, 4 mg mL⁻¹), FDH (16 μM, 0.68 mg mL⁻¹), NAD⁺ (1 mM). After 48 h, a > 99% conversion was obtained for **8a** and a 63% was obtained for **9a**. For the latter reaction, additional aliquots of FDH (8 μM) and NAD⁺ (1 mM) were added and the reaction was incubated for another 48 h, thus obtaining 97% conversion. The reactions were acidified with concentrated HCl (6 mL) and unreacted starting material was extracted with MTBE (2 × 80 mL). After the addition of KOH (10 M, 12 mL), the amine products were extracted with MTBE (2 × 120 mL). The combined organic layers were dried over MgSO₄ and concentrated under reduced pressure. Avoiding any further purification step, (*R*)-**8b** and (*R*)-**9b** were obtained with 82% (colourless liquid, 495 mg, 4 mmol) and 65% (clear, yellow liquid, 437 mg, 3 mmol) isolated yields. The purity of the products was analysed by ¹H-NMR (400 MHz in CDCl₃) (Supplementary Figs. 19 and 20) and GC equipped with chiral column (Supplementary Tables 2 and 24, and Supplementary Methods).

**In silico modifications**. All in silico modifications were performed using Yasara as software[52], and selecting AMBER 03 as force field[53]. Prior to the introduction of any in silico mutation into the model structure, the protonation state of all the atoms was corrected automatically with the exception of the atoms involved in the hydride shift from the cofactor to the substrate. The protonation state of these atoms was corrected manually upon visual inspection. After the introduction of a mutation, the energy of the system was minimised following a three-step protocol that enables the adjustment of the model structure without creating any possible unwanted deformation. In step one, only the atoms constituting the mutated amino acid residue were subjected to energy minimisation. In step two, the process for energy minimisation was repeated by including all the atoms of the amino acid residues that are located within 6 Å distance from the mutated residue. In step three, the energy of the overall structure was minimised.

**Homology modelling generation**. The homology model was created as a hybrid model and using multiple templates as specified in the Supplementary Table 12[54]. Details on the procedure are reported in Supplementary Methods.

**Molecular docking**. The molecular dockings were performed using Autodock Vina[55], both as standalone engine or as tool incorporated into Yasara. The pre-selection and analysis of the obtained docked conformations were executed using the standard Vina's scoring function implemented into in-house-scripts. This analysis resulted in a number of selected docking poses that were finally inspected visually. The refined analysis resulted in the selection of the reactive docking poses, which were used for the molecular dynamics (MD) simulations. Details on the procedure are reported in Supplementary Methods.

**Molecular dynamics simulations**. The MD simulations were performed using Yasara as software[52], and selecting AMBER 03 as force field[53]. MD simulations were executed considering both the pro-*R* and pro-*S* binding conformations of the three iminium substrate intermediates (**3c**, **4c** and **8c**). The following parameters were selected for the MD simulations: (1) the minimum number of independent simulations with random initial velocities was 6; (2) the time for each simulation was 500 ps; (3) analysis was performed by taking one snapshot every 6.25 ps. Thus, 81 frames (counting also the starting structure) were collected for each simulation and they were analysed considering different dynamic properties as explained in more details in the Supplementary Methods. Among all the inspected properties, the most important ones resulted to be: (1) the distance between the departing hydride of the coenzyme NADH and the prochiral carbon of the iminium intermediate; (2) a dihedral angle ($\chi$) that allows us to define the stereoselective outcome of the reaction based on the binding mode of the substrate in the active site[19]. In the case of the first parameter, the threshold distance for a productive hydride shift was set to the sum between the van der Walls radii of the carbon and hydrogen atoms, which is equal to ca. 3.0 Å $\left(r_H^{vdw} = 1.2 \text{ Å}, r_C^{vdw} = 1.7 \text{ Å}; r_H^{vdw} + r_C^{vdw} = 2.9 \text{ Å}; \text{thus}: d_{CH}^{threshold} = 3.0 \text{ Å}\right)$[56]. In the case of the second parameter, the dihedral angle was defined using the Cahn–Ingold–Prelog priority and considering the hydride of the NADH coenzyme along with the three atoms of the iminium substrates that are bound to the prochiral carbon. In our model system, an angle of $63.88 \pm 0.21°$ leads to the formation of the *R*-configured amine product, whereas an angle of $-63.97 \pm 0.29°$ leads to the formation of the *S*-configured amine product. A detailed description is reported in Supplementary Methods.

**Determination of NADH saturation for steady-state kinetics**. All kinetic experiments were conducted using a Shimadzu UV-1800 UV-vis spectrophotometer in HCOONH$_4$/NH$_3$ buffer (4 M, pH 9) and following the oxidation of NADH at $\lambda = 360$ nm ($\varepsilon = 4250$ M$^{-1}$ cm$^{-1}$) for 1 min ($\lambda$ of 360 nm was selected due to an overlap of the absorption of **8a** and NADH at the most frequently selected $\lambda$ of 340 nm).

The saturation concentration of NADH for Ch1-AmDH and LE-AmDH-v1 was determined with **8a** as substrate. Measurements at 60 °C were performed in two sets of independent experiments and using two different charges of purified protein.

A fixed concentration of **8a** (25 mM; from 500 mM main stock in DMSO which was further diluted to 350 mM in buffer) was added to a preincubated buffer (70 °C) and further incubated at 60 °C for 2 min in the thermostatic-controlled cuvette holder of the UV-vis spectrophotometer. The enzyme was added (2.0 μM for Ch1-AmDH and 5.2 μM for LE-AmDH-v1) and the solution was incubated for a further 20 s at the desired temperature. Then, the reaction was initiated by the addition of coenzyme (0–270 μM; from 50 mM main stock in 50 mM KPi buffer at pH 8.0, which was further diluted to 10 mM in a reaction buffer).

The initial velocities were calculated from the linear range of the fitted trend line of the progress curve and were then plotted against the NADH concentration to obtain the $K_M$ values (Supplementary Methods and Supplementary Table 18). The Michaelis-Menten equation fitted well with the experimental determination.

**Steady-state kinetics**. Two sets of independent measurements were performed using two different charges of purified protein. Varied substrate concentrations (from 500 mM main stock in DMSO, which was further diluted to 350 mM in buffer) were added to a preincubated buffer (with an incubation temperature 10 °C higher than the measuring temperature) and further incubated at the desired measuring temperature in the thermostatic controlled cuvette holder of the UV-vis spectrophotometer for 2 min. The enzyme was added and the solution was incubated for a further 20 s, after which the reaction was initiated by the addition of NADH (250 μM; from 50 mM main stock in 50 mM KPi buffer at pH 8.0, which was further diluted to 10 mM in a reaction buffer).

The initial velocities were calculated from the linear range of the fitted trend line of the progress curve and were then plotted against the substrate concentration to obtain the kinetic parameter.

A summary of the reaction conditions used for the assay is provided in the Supplementary Methods and Supplementary Table 19. The results are summarised in Table 2. The Michaelis–Menten equation fitted well with the experimental determinations.

**Determination of the IC$_{50}$**[15]. Product inhibition studies were performed in duplicate at 60 °C. A fixed concentration of **8a** (15 mM; from 500 mM main stock in DMSO, which was further diluted to 350 mM in buffer) and varied concentrations of product (*R*)-**8b**, (inhibitor, 0–25 mM; from 500 mM main stock in DMSO, which was further diluted to 350 mM in buffer) were added to a pre-incubated buffer (70 °C) and further incubated at 60 °C for 2 min in the thermostatic-controlled cuvette holder of the UV-vis spectrophotometer. The enzyme was added (2.1–10.7 μM for Ch1-AmDH and 4.9 μM for LE-AmDH-v1) and the solution was incubated at the desired temperature for a further 30 s, after which the reaction was initiated by the addition of coenzyme (250 μM; from 50 mM main stock in a 50 mM KPi buffer at pH 8.0, which was further diluted to 10 mM in a reaction buffer).

The initial velocities were calculated from the linear range of the fitted trend line of the progress curve and were then plotted against the inhibitor concentration to obtain IC$_{50}$ at a fixed substrate concentration (this value represents the amount of amine product that correlates with a decrease of 50% of the initial catalytic activity due to product inhibition at 15 mM **8a**) and is displayed in Fig. 4b. For details, see Supplementary Amine Methods and Supplementary Table 20.

**Determination of the $K_I$ for (*R*)-8b**. Varied concentrations of substrate **8a** (0–30 mM, from 500 mM main stock in DMSO, which was further diluted to 350 mM in buffer) and inhibitor (*R*)-**8b** (0, 5 and 10 mM for LE-AmDH-v1 and 0, 0.2, and 0.4 mM for Ch1-AmDH, from 500 mM main stock in DMSO, which was further diluted to 350 mM in buffer) were added to a preincubated buffer (70 °C) and further incubated at 60 °C for 2 min in the thermostatic controlled cuvette holder of the UV-vis spectrophotometer. The enzyme was added (4.5–9 μM for LE-AmDH-v1 and 2.9–6.4 μM for Ch1-AmDH) and the solution was incubated for a further 30 s, after which the reaction was initiated by the addition of NADH (250 μM; from 50 mM main stock in 50 mM KPi buffer at pH 8.0, which was further diluted to 10 mM in a reaction buffer).

The initial velocities were calculated from the linear range of the fitted trend line. Primary double reciprocal Lineweaver-Burk plots were generated for each set of experiments at different inhibitor concentrations (Fig. 4c, d). The slopes obtained from these primary plots were plotted against the inhibitor concentration to obtain $K_I$ (Fig. 4e)[43]. $K_M^{eff}$ was determined according to the equation described in Supplementary Methods[41], and the results are summarised in Table 2 and Supplementary Tables 21–23.

**Reporting summary**. Further information on research design is available in the Nature Research Reporting Summary linked to this article.

## Data availability

The raw data underlying Figs. 3, 4, 5, Supplementary Methods-Supplementary Figs. 24, 36–41, and Supplementary Methods-Supplementary Table 17 are provided as a Source Data file. Note that Table 2 is generated from the same source data of Supplementary Methods-Supplementary Figs. 37–41, whereas Supplementary Methods-Supplementary Tables 18 and 21 are generated from the same source data of Supplementary Methods-Supplementary Figs. 37 and 41, respectively. Figure 4e and Supplementary Methods-Supplementary Fig. 42 consist in a graphical/mathematical elaboration of Source Data derived from Supplementary Methods-Supplementary Fig. 41 and following the procedure described in ref. [43]. All other data are available from the corresponding authors upon reasonable request.

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

## Acknowledgements

This project has received funding from the European Research Council (ERC) under the European Union's Horizon 2020 Research and Innovation programme (ERC-StG, Grant Agreement No. 638271, BioSusAmin). Dutch funding from the NWO Sector Plan for Physics and Chemistry is also acknowledged.

## Author contributions

All of the authors contributed to the design of the experiments. V.T. expressed and purified the enzymes, performed and analysed the biocatalytic reactions by GC, optimised the reaction conditions, performed thermal stability measurements, and performed preparative scale amination. T.K. performed the tasks of molecular biology for the creation of the gene libraries of variants, as well as the kinetics, inhibition, and long-term stability studies, and analysed the kinetic data. M.F.M. performed the homology modelling, the molecular modelling and the molecular dynamics simulations. V.T., M.F.M. and F.G.M. designed the focused library of variants. M.L.C. performed the chemical synthesis and identification of 6-oxo carboxylic acids and their methyl esters, 6-amino carboxylic acids methyl esters, and analysed biocatalytic reactions by RP-HPLC. F.G.M. and T.K. directed the project. F.G.M. conceived the project and received the funding. V.T., T.K., M.F.M. and F.G.M. prepared the paper and Supplementary Information, which were revised and approved by all the authors.

## Additional information

**Competing interests:** The authors declare no competing interests.

