## [Peer Review File · Nature Communications]

REVIEWERS' COMMENTS:

Reviewer #1 (Remarks to the Author):

I went through the responses and checked the manuscript. I feel that the manuscript can be accepted for publication. I look forward to reading the published manuscript.

Reviewer #3 (Remarks to the Author):

In the very careful revision process of the manuscript NCOMMS-19-13989-T, Knaus and Mutti and co-authors succeeded in improving their very nice manuscript further by addressing thoroughly the reviewers' comments.

In particular, I think that the added new synthetic protocols including a work-up and the reported isolated yields for these experiments (thus, addressing the report of reviewer #3, issue (1) therein) will be helpful for the reader. For example, organic chemists being interested in using enzymes for preparative purpose will appreciate such an information, which I think is very valuable.

In addition, according to my opinion also the direct comparison with the performance of previous biotransformations (thus, addressing the report of reviewer #3, issue (5) therein) now gives a better insight into the high potential of the new enzyme developed by Knaus and Mutti and co-workers compared to the more general statement in the previous paragraph.

In conclusion, I recommend this carefully revised version of this very nice manuscript NCOMMS-19-13989-T for publication in *Nature Communications*.

One recommendation, however, I still have which is related to the answer of Knaus and Mutti and co-workers to the report of reviewer #3, issue (8): first, I am fully satisfied with the answer of the authors given in the first section of their reply (that using mg is sufficient in case of purified enzymes). Second, I also found the last part of their answer interesting related to the "unit per mg"-data, which is in the second text section ("... for the sake of completeness, we can say to Reviewer #3 that the activity of LE-AmDH-v1 was in general ca. 730 U mg⁻¹ during our study depending on the batch (measured for benzaldehyde at 60 °C in the ammonium buffer 2 M, pH 9.0). ...") and I was wondering if at least this statement given in parentheses on the unit per mg of purified enzyme for benzaldehyde under these conditions might be added to the Supporting Information as I assume that it could be interesting for those readers who prefer to read "unit per mg"-data when it comes to the characterization of enzymes.

Reviewer #4 (Remarks to the Author):

The Manuscript submitted by Mutti describes the generation of catalysts for the NADH-dependent R-selective reductive amination of ketones. The efficient preparation of chiral amines is an important field, and lots of research in the past 10 years afforded a growing amount of different enzymes for the biocatalytic preparation of chiral amines.

Novelty: Amine dehydrogenases for reductive amination of ketones are not observed in nature and thus have to be generated by protein engineering. The first breakthrough (evolved amine dehydrogenase for R-selective ketone amination from a leucine dehydrogenase) was published 2012 in *Angewandte Chemie* by the Bommarius group. In the meantime, a handful of other amine dehydrogenases have been created by protein engineering starting from other amino acid

dehydrogenases.

The present work created amine dehydrogenases for the R-selective reductive amination of ketones with improved properties: A range of previously non-accessible ketones (bulky arylalkyl ketones, chromanones) are now accessible, and the here-presented amine dehydrogenases show a significantly lower product inhibition, resulting in a 15-fold higher catalyst efficiency compared to known amine dehydrogenases. A further benefit is its improved stability compared to previously described AmDH.

The here-presented protein engineering can therefore be seen as an optimization and broadening of an existing biocatalytic approach by providing enzymes generated from another protein as starting point (lysine-6-dehydrogenase) compared to previous protein engineering studies, which is an interesting variation.

A possible acceptance of the paper in this journal should thus be judged on the novelty and innovation of the above-described achievements.

Overall, the study is based on sound experiments (biocatalysis, kinetic characterization, preparative reactions, modeling studies; the manuscript improved sufficiently by consideration of previous referee's comments), and the achievements are for sure important for the field, but personally, I don't see the high innovation and transferable knowledge increase required for a Nature Communication publication.

Before accepting this manuscript elsewhere, there are a lot of issues in the language and argumentation, which should be adjusted.

Title:

The term "unprecedented" is very unspecific. All research results should be new and thus will give unprecedented results / catalysts / Related to the content of the manuscript, "unprecedented" mainly refers to a 15-fold improved efficiency and broadening of the substrate scope. There are few examples in literature where protein engineering created biocatalysts with dramatically improved properties such as activity and stability yielding catalysts suitable for industrial applications (e.g. the codexis sitagliptin transaminase). For creating an enzyme with broadened substrate scope and "unprecedented" properties, I would expect activities at least in the range or above wild-type activities for the natural substrates. I guess that this is by far not reached in this study as enzyme concentrations of 90 μM corresponding to 3-4 mg/mL (purified enzyme!) has to be used (Please also mention protein concentration in mg/mL in the text). Please add a comparison (specific activity or kcat/Km) of lysine oxidation of the wild type versus 1-phenylethylamine oxidation and acetophenone amination by the best variant in the manuscript to give a clear orientation of the catalytic power of the variant.

With the term "Cryptostereoselective scaffold" the authors want to describe the fact that the wildtype enzyme does not introduce a chiral center in its natural reaction, but the enzyme shows stereoselectivity if a prochiral unnatural substrate is employed. The term "stereoselective" has a mathematical definition, and always refers to a given substrate. Enzymatic stereoselectivity can be measured for different substrates and thus expressed as distinct values. "Cryptostereoselectivity" cannot be measured. Cryptostereoselective just means "not-yet-known" stereoselective for a not-yet-investigated substrate. Therefore, in my opinion, this term is not very useful (despite sounding fancy), especially if it is referred to a "scaffold" as suggested in the title. I strongly recommend to delete this definition in the manuscript. The circumstance that the other reviewers have not criticized this definition previously does not imply that they find it useful / acceptable.

Line 25: "not exhibit any stereoselectivity in its natural reaction". This is not true in this generalized sense, as the enzyme only converts L- but not D-lysine. This is also a kind of stereoselectivity! The same issue has the sentence in the conclusion (line 404-405).

Line 137-139 "information on the reaction's stereoselectivity could not be obtained" → this makes no sense, because this reaction will never be stereoselective because of the achiral product generated in this reaction.

Line 142 "this experiment would reveal the crypto-stereoselectivity of LysEDH" → would be much clearer if it would read "this experiment would reveal a possible stereoselectivity of LysEDH in the conversion of 3a". I agree that it was previously not known whether the enzyme converts ketone substrates such as 3a, and thus this is a clever and interesting question investigated by the authors, but its scientific value will still exist without this confusion term.

Other language problems in the main text:

Abstract:

Line 24: "Unconventional enzyme scaffold" – what is the meaning of "unconventional" related to scaffold here? The sentence would be less confusing if "unconventional" is deleted.

Please rephrase lines 30-31: "can be a critical factor" is not giving useful information here in the abstract. You could mention that albeit a lower k_{cat} compared to previously existing AmDH, the generated variant shows a x-fold lower product inhibition in the amination of acetophenone, and this generates catalysts suitable for preparative scale applications as the conversions at 100 mM substrate load are improved to...%.

Introduction:

Line 48-50: MAO are not applied at industrial scale for deracemization, as far as I know. The only example I know is a desymmetrization. (J Am Chem Soc 2012, 134:6467- 6472)

Results/Conclusion:

Line 353: "Enantioselective preferences" must read "enantioference". (A preference cannot be enantioselective).

Line 358: "A stereo-switchable selectivity" must read "a substrate-dependent switch of the enantioference". (Please also correct this in lines 26, 361, 364).

The authors are arguing that this substrate-dependent reversal of the enantioference is a unique feature, suggesting its importance. I agree that this behavior of this enzyme is very interesting, but in my point of view, it is a pity that this is happening here, because it would be much more valuable to create an amine dehydrogenase with S-selectivity towards ketone amination (complementary to the existing R-selective ones!). I guess that the creation of such an S-selective AmDH was the expectation when starting with especially this scaffold, as product 2b is indeed an S-amine. Therefore I encourage the authors to mention this, and also would rather focus on explaining the mechanism of the stereo inversion, which is already explained shortly in lines 397-400. Compared to other amine dehydrogenases that required a redesign of the residues involved in substrate's carboxylate binding, the engineering of the here-generated variants did not required to touch these residues, interestingly. This is possible by creating the space by the F173A substitution and allowing non-acidic substrate to bind in the alternative binding mode (but at the expense of the inverted stereopreference).

Line 368: "Molecular docking simulations" should read "molecular docking and molecular dynamics simulations"

Line 481: "non-bonding dihedral angle" makes no sense to me. There are non-bonding atoms in a simulation, but neither "bonding" nor "non-bonding" dihedrals exist. Suggestion: just delete "non-bonding". The authors performed a technical measurement of the dihedral defined by the 4 atoms relevant in the CIP-nomenclature.

In summary, after addressing the above-mentioned comments, I believe that the manuscript is suitable for publication in a more specialized journal.

Reply to Reviewer's Comments:

Reviewer #1 (Remarks to the Author):

I went through the responses and checked the manuscript. I feel that the manuscript can be accepted for publication. I look forward to reading the published manuscript.

Reviewer #3 (Remarks to the Author):

In the very careful revision process of the manuscript NCOMMS-19-13989-T, Knaus and Mutti and co-authors succeeded in improving their very nice manuscript further by addressing thoroughly the reviewers' comments. In particular, I think that the added new synthetic protocols including a work-up and the reported isolated yields for these experiments (thus, addressing the report of reviewer #3, issue (1) therein) will be helpful for the reader. For example, organic chemists being interested in using enzymes for preparative purpose will appreciate such an information, which I think is very valuable.

In addition, according to my opinion also the direct comparison with the performance of previous biotransformations (thus, addressing the report of reviewer #3, issue (5) therein) now gives a better insight into the high potential of the new enzyme developed by Knaus and Mutti and co-workers compared to the more general statement in the previous paragraph.

In conclusion, I recommend this carefully revised version of this very nice manuscript NCOMMS-19-13989-T for publication in Nature Communications.

One recommendation, however, I still have which is related to the answer of Knaus and Mutti and co-workers to the report of reviewer #3, issue (8): first, I am fully satisfied with the answer of the authors given in the first section of their reply (that using mg is sufficient in case of purified enzymes). Second, I also found the last part of their answer interesting related to the "unit per mg"-data, which is in the second text section ("... for the sake of completeness, we can say to Reviewer #3 that the activity of LE-AmDH-v1 was in general ca. 730 mU mg⁻¹ during our study depending on the batch (measured for benzaldehyde at 60 °C in the ammonium buffer 2 M, pH 9.0). ...") and I was wondering if at least this statement given in parentheses on the unit per mg of purified enzyme for benzaldehyde under these conditions might be added to the Supporting Information as I assume that it could be interesting for those readers who prefer to read "unit per mg"-data when it comes to the characterization of enzymes.

This additional information—provided in the previous Reply to Reviewer's Comments—was also added to the Supplementary Information as Supplementary methods in the sub-heading "Protein expression, purification and activity assays".

Reviewer #4 (Remarks to the Author):

The Manuscript submitted by Mutti describes the generation of catalysts for the NADH-dependent R-selective reductive amination of ketones. The efficient preparation of chiral amines is an important field, and lots of research in the past 10 years afforded a growing amount of different enzymes for the biocatalytic preparation of chiral amines.

Novelty: Amine dehydrogenases for reductive amination of ketones are not observed in nature and thus have to be generated by protein engineering. The first breakthrough (evolved amine dehydrogenase for R-selective ketone amination from a leucine dehydrogenase) was published 2012 in Angewandte Chemie by the Bommarius group. In the meantime, a handful of other amine dehydrogenases have been created by protein engineering starting from other amino acid dehydrogenases.

The present work created amine dehydrogenases for the R-selective reductive amination of ketones with improved properties: A range of previously non-accessible ketones (bulky arylalkyl ketones, chromanones) are

now accessible, and the here-presented amine dehydrogenases show a significantly lower product inhibition, resulting in a 15-fold higher catalyst efficiency compared to known amine dehydrogenases. A further benefit is its improved stability compared to previously described AmDH.

The here-presented protein engineering can therefore be seen as an optimization and broadening of an existing biocatalytic approach by providing enzymes generated from another protein as starting point (lysine-6-dehydrogenase) compared to previous protein engineering studies, which is an interesting variation.

A possible acceptance of the paper in this journal should thus be judged on the novelty and innovation of the above-described achievements.

Overall, the study is based on sound experiments (biocatalysis, kinetic characterization, preparative reactions, modeling studies; the manuscript improved sufficiently by consideration of previous referee's comments), and the achievements are for sure important for the field, but personally, I don't see the high innovation and transferable knowledge increase required for a Nature Communication publication.

Before accepting this manuscript elsewhere, there are a lot of issues in the language and argumentation, which should be adjusted.

Title:

The term "unprecedented" is very unspecific. All research results should be new and thus will give unprecedented results / catalysts / ... Related to the content of the manuscript, "unprecedented" mainly refers to a 15-fold improved efficiency and broadening of the substrate scope. There are few examples in literature where protein engineering created biocatalysts with dramatically improved properties such as activity and stability yielding catalysts suitable for industrial applications (e.g. the codexis sitagliptin transaminase). For creating an enzyme with broadened substrate scope and "unprecedented" properties, I would expect activities at least in the range or above wild-type activities for the natural substrates. I guess that this is by far not reached in this study as enzyme concentrations of 90 μM corresponding to 3-4 mg/mL (purified enzyme!) has to be used (Please also mention protein concentration in mg/mL in the text). Please add a comparison (specific activity or $k_{\text{cat}}/K_{\text{m}}$) of lysine oxidation of the wild type versus 1-phenylethylamine oxidation and acetophenone amination by the best variant in the manuscript to give a clear orientation of the catalytic power of the variant.

- The term "unprecedented" was removed from the title and from the manuscript to comply with the Language and Style guidelines of Nature Communications. The title is now "Generation of Amine Dehydrogenases with Increased Catalytic Performance and Substrate Scope from ϵ -deaminating Lysine Dehydrogenase".

- The amount of enzyme in mass was added in the text in the Methods section.

- In truth, we do not think that a comparison between the specific activity for L-lysine oxidation catalysed by the wild-type (ϵ -deaminating) lysine dehydrogenase versus the specific activity for 1-phenylethylamine oxidation and acetophenone amination catalysed by the best variant in the manuscript would give a clear orientation of the catalytic power of the variant. The catalytic power of the variant was already illustrated extensively by the kinetic and inhibition studies, which provide an objective description regardless on the rate magnitude of the natural reaction towards the natural substrate.

However, the following data were added as Supplementary Methods following the comment by Reviewer #4 and also the previous remark by Reviewer #3:

- The specific activity of the wild-type (ϵ -deaminating) L-lysine dehydrogenase towards L-lysine is known as it was determined by Heydari et al. in their original publication (ref. 34, *Appl. Environ. Microbiol.* **2004**, 937-942 (2004)). The reported value is 7.81 U mg^{-1} [measured for L-lysine (10 mM) in glycine-KOH buffer (100 mM, pH 10), at 50 °C].

- The catalytic activity for the amination of benzaldehyde catalysed by LE-AmDH-v1 was in general ca. 730 mU mg^{-1} during our study depending on the batch [measured for benzaldehyde (10 mM) in the ammonium

buffer (2 M, pH 9.0) at 60 °C]. This assay was used for a practical and rapid comparison among the enzyme batches produced in this study.

- The catalytic activity for the amination of acetophenone catalysed by LE-AmDH-v1 was in general ca. 200 mU mg⁻¹ [measured for acetophenone (30 mM) in the ammonium buffer (2 M, pH 9.0) at 60 °C].

With the term “Cryptostereoselective scaffold” the authors want to describe the fact that the wildtype enzyme does not introduce a chiral center in its natural reaction, but the enzyme shows stereoselectivity if a prochiral unnatural substrate is employed. The term “stereoselective” has a mathematical definition, and always refers to a given substrate. Enzymatic stereoselectivity can be measured for different substrates and thus expressed as distinct values. “Cryptostereoselectivity” cannot be measured. Cryptostereoselective just means “not-yet-known” stereoselective for a not-yet-investigated substrate. Therefore, in my opinion, this term is not very useful (despite sounding fancy), especially if it is referred to a “scaffold” as suggested in the title. I strongly recommend to delete this definition in the manuscript. The circumstance that the other reviewers have not criticized this definition previously does not imply that they find it useful/acceptable.

We accept the criticism of Reviewer #4. Therefore, the term cryptostereoselective was removed from the title and the manuscript. The related paragraph in the introduction was also removed.

Line 25: “not exhibit any stereoselectivity in its natural reaction”. This is not true in this generalized sense, as the enzyme only converts L- but not D-lysine. This is also a kind of stereoselectivity! The same issue has the sentence in the conclusion (line 404-405).

The sentences in the abstract and in the conclusion were modified as follows: “...does not operate any asymmetric transformation in its natural reaction”. In other words, the wild-type enzyme recognises L-lysine over D-lysine but the natural reaction does not install a new stereogenic centre.

Line 137-139 “information on the reaction’s stereoselectivity could not be obtained” → this makes no sense, because this reaction will never be stereoselective because of the achiral product generated in this reaction.

From the comment of the reviewer, we realised that unfortunate choice of the verb. The sentence was changed as follows: “As **2b** is a terminal (achiral) amine product, information on the reaction’s stereoselectivity was not attainable”. In other words, it was not possible to obtain such information because of the achiral product generated in this reaction.

Line 142 “this experiment would reveal the crypto-stereoselectivity of LysEDH” → would be much clearer if it would read “this experiment would reveal a possible stereoselectivity of LysEDH in the conversion of 3a”. I agree that it was previously not known whether the enzyme converts ketone substrates such as 3a, and thus this is a clever and interesting question investigated by the authors, but its scientific value will still exist without this confusion term.

The text was changed accordingly to reviewer’s comment.

Other language problems in the main text:

Abstract:

Line 24: “Unconventional enzyme scaffold” – what is the meaning of “unconventional” related to scaffold here? The sentence would be less confusing if “unconventional” is deleted.

The sense of the word “unconventional” is explained in the introduction of the manuscript. The use of “unconventional” in the abstract relates to this concept: “..... Conversely, known engineered AmDHs have been obtained starting exclusively from L-amino acid dehydrogenases and by mutating the same highly conserved amino acid residues in the active site (i.e., lysine and aspartate) Ref. 27-31”.

In other words—with unconventional scaffold—we want to point out that we have chosen a strategy for engineering and a related scaffold that are different from the more obvious (i.e., *conventional*) choice of starting from L- amino acid dehydrogenases that deaminates at the α -amino position, as it was done in all the published studies so far.

For this reason, the word “unconventional” was retained in this revised version.

Please rephrase lines 30-31: “can be a critical factor” is not giving useful information here in the abstract. You could mention that albeit a lower kcat compared to previously existing AmDH, the generated variant shows a x-fold lower product inhibition in the amination of acetophenone, and this generates catalysts suitable for preparative scale applications as the conversions at 100 mM substrate load are improved to...%.

We understand this comment from the Reviewer #4, but we point out that the Nature Communications allows for a maximum of approximately 150 words for the abstract. The abstract counts already 151 words. The text suggested by the reviewer would add other 43 words, making the abstract unacceptable by the Editorial office of the journal.

In truth, it was a difficult task to write an effective and comprehensive abstract while respecting the word count fixed by the journal. As in our opinion the abstract could not be improved further, the abstract was not modified.

Introduction:

Line 48-50: MAO are not applied at industrial scale for deracemization, as far as I know. The only example I know is a desymmetrization. (J Am Chem Soc 2012, 134:6467- 6472)

We assume that Reviewer #4 refers specifically to the “*industrial scale*” because there are many examples of MAOs applied in deracemization. Different companies supply and/or utilise MAO biocatalysts for kinetic resolution and deracemization, especially, pharmaceutical industries. As it can be that these processes are not so popular at the moment, the sentence was slightly revised according to the concern of Reviewer #4: “*Classical industrially-applied and laboratory scale biocatalytic methods convert ...*”.

As examples of deracemization involving MAO enzymes, we mention here a few articles from Turner’s group, in which the titles are themselves elucidating in this regards:

- 1) O’Reilly, E., Iglesias, C. & Turner, N. J. Monoamine Oxidase- ω -Transaminase Cascade for the Deracemisation and Dealkylation of Amines. *ChemCatChem* 6, 992-995, (2014).
- 2) Foulkes, J. M., Malone, K. J., Coker, V. S., Turner, N. J. & Lloyd, J. R. Engineering a Biometallic Whole Cell Catalyst for Enantioselective Deracemization Reactions. *ACS Catal.* 1, 1589-1594, (2011)
- 3) Alexeeva, M., Enright, A., Dawson, M. J., Mahmoudian, M. & Turner, N. J. Deracemization of Methylbenzylamine Using an Enzyme Obtained by In Vitro Evolution. *Angew. Chem.* 114, 3309-3312, (2002).
- 4) Schrittwieser, J. H. *et al.* Deracemization by simultaneous bio-oxidative kinetic resolution and stereoinversion. *Angew. Chem. Int. Ed.* 53, 3731-3734, (2014).
- 5) Heath, R. S., Pontini, M., Hussain, S. & Turner, N. J. Combined Imine Reductase and Amine Oxidase Catalyzed Deracemization of Nitrogen Heterocycles. *ChemCatChem* 8, 117-120, (2016).

Results/Conclusion:

Line 353: “Enantioselective preferences” must read “enantiopreference”. (A preference cannot be enantioselective).

It was changed accordingly.

Line 358: "A stereo-switchable selectivity" must read "a substrate-dependent switch of the enantiopreference". (Please also correct this in lines 26, 361, 364).

Thanks for the suggestion. It was changed in all cases with a slight modification from our side: "A substrate-dependent switch of enantioselectivity..."

The authors are arguing that this substrate-dependent reversal of the enantiopreference is a unique feature, suggesting its importance. I agree that this behavior of this enzyme is very interesting, but in my point of view, it is a pity that this is happening here, because it would be much more valuable to create an amine dehydrogenase with S-selectivity towards ketone amination (complementary to the existing R-selective ones!). I guess that the creation of such an S-selective AmDH was the expectation when starting with especially this scaffold, as product 2b is indeed an S-amine. Therefore I encourage the authors to mention this, and also would rather focus on explaining the mechanism of the stereo inversion, which is already explained shortly in lines 397-400. Compared to other amine dehydrogenases that required a redesign of the residues involved in substrate's carboxylate binding, the engineering of the here-generated variants did not required to touch these residues, interestingly. This is possible by creating the space by the F173A substitution and allowing non-acidic substrate to bind in the alternative binding mode (but at the expense of the inverted stereopreference).

To refer to his previous comment, herein, Reviewer #4 has well-summarised the "unconventional" properties of our strategy and choice of scaffold.

Going to the specific part of the comment, in truth, our aim was (and still is) to obtain a library of stereocomplementary variants that are capable of producing pharmaceutically relevant both R- and S-configured amines in perfect optical purity. In this initial work, we succeeded to obtain useful R-selective AmDHs. Current work is focused on obtaining S-selective AmDHs from the same scaffold.

Finally, we decided to not dedicate ample space to the discussion of the substrate-dependent switch of enantioselectivity because our work was based on a homology model and not on a X-ray crystal structure. Therefore, we decided to avoid excessive speculation at this stage. By solving the crystal structure in future, we will be able to elucidate properly this point. Furthermore, the maximum allowed length of papers by journal policy would not allow ample discussion of this aspect in this manuscript.

Line 368: "Molecular docking simulations" should read "molecular docking and molecular dynamics simulations"

We agree with Reviewer #4 at this point. The change was done accordingly.

Line 481: "non-bonding dihedral angle" makes no sense to me. There are non-bonding atoms in a simulation, but neither "bonding" nor "non-bonding" dihedrals exist. Suggestion: just delete "non-bonding". The authors performed a technical measurement of the dihedral defined by the 4 atoms relevant in the CIP-nomenclature.

After some re-reading and searching for the latest IUPAC nomenclatures (see note [a]), we agree with deleting the term "non-bonding". However, it is not for the reasoning that was given by Reviewer #4, but because it seems redundant according to the IUPAC suggestions.

Ref:

[a] IUPAC. Compendium of Chemical Terminology, 2nd ed. (the "Gold Book"). Compiled by A. D. McNaught and A. Wilkinson. Blackwell Scientific Publications, Oxford (1997). XML on-line corrected version:

<http://goldbook.iupac.org> (2006-) created by M. Nic, J. Jirat, B. Kosata; updates compiled by A. Jenkins. ISBN 0-9678550-9-8. <https://doi.org/10.1351/goldbook>.

Last update: 2014-02-24; version: 2.3.3.

DOI of this term: <https://doi.org/10.1351/goldbook.D01730>.

Original PDF version: <http://www.iupac.org/goldbook/D01730.pdf>.